# Accelerated Mini-Batch Stochastic Dual Coordinate Ascent

**Shai Shalev-Shwartz**
School of Computer Science and Engineering
Hebrew University, Jerusalem, Israel

**Tong Zhang**
Department of Statistics
Rutgers University, NJ, USA

## Abstract

Stochastic dual coordinate ascent (SDCA) is an effective technique for solving regularized loss minimization problems in machine learning. This paper considers an extension of SDCA under the mini-batch setting that is often used in practice. Our main contribution is to introduce an accelerated mini-batch version of SDCA and prove a fast convergence rate for this method. We discuss an implementation of our method over a parallel computing system, and compare the results to both the vanilla stochastic dual coordinate ascent and to the accelerated deterministic gradient descent method of Nesterov [2007].

## 1  Introduction

We consider the following generic optimization problem. Let $\phi_1, \ldots, \phi_n$ be a sequence of vector convex functions from $\mathbb{R}^d$ to $\mathbb{R}$, and let $g : \mathbb{R}^d \to \mathbb{R}$ be a strongly convex regularization function. Our goal is to solve $\min_{x \in \mathbb{R}^d} P(x)$ where

$$P(x) = \left[ \frac{1}{n} \sum_{i=1}^{n} \phi_i(x) + g(x) \right]. \tag{1}$$

For example, given a sequence of $n$ training examples $(v_1, y_1), \ldots, (v_n, y_n)$, where $v_i \in \mathbb{R}^d$ and $y_i \in \mathbb{R}$, ridge regression is obtained by setting $g(x) = \frac{\lambda}{2}\|x\|^2$ and $\phi_i(x) = (x^\top v_i - y_i)^2$. Regularized logistic regression is obtained by setting $\phi_i(x) = \log(1 + \exp(-y_i x^\top v_i))$.

The *dual* problem of (1) is defined as follows: For each $i$, let $\phi_i^* : \mathbb{R}^d \to \mathbb{R}$ be the convex conjugate of $\phi_i$, namely, $\phi_i^*(u) = \max_{z \in \mathbb{R}^d}(z^\top u - \phi_i(z))$. Similarly, let $g^*$ be the convex conjugate of $g$. The dual problem is:

$$\max_{\alpha \in \mathbb{R}^{d \times n}} D(\alpha) \quad \text{where} \quad D(\alpha) = \left[ \frac{1}{n} \sum_{i=1}^{n} -\phi_i^*(-\alpha_i) - g^*\left( \tfrac{1}{n} \sum_{i=1}^{n} \alpha_i \right) \right], \tag{2}$$

where for each $i$, $\alpha_i$ is the $i$'th column of the matrix $\alpha$.

The dual objective has a different dual vector associated with each primal function. Dual Coordinate Ascent (DCA) methods solve the dual problem iteratively, where at each iteration of DCA, the dual objective is optimized with respect to a single dual vector, while the rest of the dual vectors are kept in tact. Recently, Shalev-Shwartz and Zhang [2013a] analyzed a stochastic version of dual coordinate ascent, abbreviated by SDCA, in which at each round we choose which dual vector to optimize uniformly at random (see also Richtárik and Takáč [2012a]). In particular, let $x^*$ be the optimum of (1). We say that a solution $x$ is $\epsilon$-accurate if $P(x) - P(x^*) \leq \epsilon$. Shalev-Shwartz and Zhang [2013a] have derived the following convergence guarantee for SDCA: If $g(x) = \frac{\lambda}{2}\|x\|_2^2$ and each $\phi_i$ is $(1/\gamma)$-smooth, then for every $\epsilon > 0$, if we run SDCA for at least

$$\left( n + \tfrac{1}{\lambda \gamma} \right) \log((n + \tfrac{1}{\lambda \gamma}) \cdot \tfrac{1}{\epsilon})$$

iterations, then the solution of the SDCA algorithm will be $\epsilon$-accurate (in expectation). This convergence rate is significantly better than the more commonly studied stochastic gradient descent (SGD) methods that are related to SDCA[1].

Another approach to solving (1) is deterministic gradient descent methods. In particular, Nesterov [2007] proposed an accelerated gradient descent (AGD) method for solving (1). Under the same conditions mentioned above, AGD finds an $\epsilon$-accurate solution after performing

$$O\left(\frac{1}{\sqrt{\lambda\gamma}}\log(\tfrac{1}{\epsilon})\right)$$

iterations.

The advantage of SDCA over AGD is that each iteration involves only a single dual vector and usually costs $O(d)$. In contrast, each iteration of AGD requires $\Omega(nd)$ operations. On the other hand, AGD has a better dependence on the condition number of the problem — the iteration bound of AGD scales with $1/\sqrt{\lambda\gamma}$ while the iteration bound of SDCA scales with $1/(\lambda\gamma)$.

In this paper we describe and analyze a new algorithm that interpolates between SDCA and AGD. At each iteration of the algorithm, we randomly pick a subset of $m$ indices from $\{1,\ldots,n\}$ and update the dual vectors corresponding to this subset. This subset is often called a mini-batch. The use of mini-batches is common with SGD optimization, and it is beneficial when the processing time of a mini-batch of size $m$ is much smaller than $m$ times the processing time of one example (mini-batch of size 1). For example, in the practical training of neural networks with SGD, one is always advised to use mini-batches because it is more efficient to perform matrix-matrix multiplications over a mini-batch than an equivalent amount of matrix-vector multiplication operations (each over a single training example). This is especially noticeable when GPU is used: in some cases the processing time of a mini-batch of size 100 may be the same as that of a mini-batch of size 10. Another typical use of mini-batch is for parallel computing, which was studied by various authors for stochastic gradient descent (e.g., Dekel et al. [2012]). This is also the application scenario we have in mind, and will be discussed in greater details in Section 3.

Recently, Takác et al. [2013] studied mini-batch variants of SDCA in the context of the Support Vector Machine (SVM) problem. They have shown that the naive mini-batching method, in which $m$ dual variables are optimized in parallel, might actually increase the number of iterations required. They then describe several "safe" mini-batching schemes, and based on the analysis of Shalev-Shwartz and Zhang [2013a], have shown several speed-up results. However, their results are for the non-smooth case and hence they do not obtain linear convergence rate. In addition, the speed-up they obtain requires some spectral properties of the training examples. We take a different approach and employ Nesterov's acceleration method, which has previously been applied to mini-batch SGD optimization. This paper shows how to achieve acceleration for SDCA in the mini-batch setting. The pseudo code of our Accelerated Mini-Batch SDCA, abbreviated by ASDCA, is presented below.

---

**Procedure Accelerated Mini-Batch SDCA**

**Parameters** scalars $\lambda, \gamma$ and $\theta \in [0,1]$ ; mini-batch size $m$
**Initialize** $\alpha_1^{(0)} = \cdots = \alpha_n^{(0)} = \bar{\alpha}^{(t)} = 0$, $x^{(0)} = 0$
**Iterate:** for $t = 1, 2, \ldots$
   $u^{(t-1)} = (1-\theta)x^{(t-1)} + \theta\nabla g^*(\bar{\alpha}^{(t-1)})$
   Randomly pick subset $I \subset \{1,\ldots,n\}$ of size $m$ and update the dual variables in $I$
      $\alpha_i^{(t)} = (1-\theta)\alpha_i^{(t-1)} - \theta\nabla\phi_i(u^{(t-1)})$ for $i \in I$
      $\alpha_j^{(t)} = \alpha_j^{(t-1)}$ for $j \notin I$
   $\bar{\alpha}^{(t)} = \bar{\alpha}^{(t-1)} + n^{-1}\sum_{i\in I}(\alpha_i^{(t)} - \alpha_i^{(t-1)})$
   $x^{(t)} = (1-\theta)x^{(t-1)} + \theta\nabla g^*(\bar{\alpha}^{(t)})$
**end**

---

In the next section we present our main result — an analysis of the number of iterations required by ASDCA. We focus on the case of Euclidean regularization, namely, $g(x) = \frac{\lambda}{2}\|x\|^2$. Analyzing more general strongly convex regularization functions is left for future work. In Section 3 we discuss

parallel implementations of ASDCA and compare it to parallel implementations of AGD and SDCA. In particular, we explain in which regimes ASDCA can be better than both AGD and SDCA. In Section 4 we present some experimental results, demonstrating how ASDCA interpolates between AGD and SDCA. The proof of our main theorem is differed to a long version of this paper (Shalev-Shwartz and Zhang [2013b]). We conclude with a discussion of our work in light of related works in Section 5.

## 2 Main Results

Our main result is a bound on the number of iterations required by ASDCA to find an $\epsilon$-accurate solution. In our analysis, we only consider the squared Euclidean norm regularization,

$$g(x) = \frac{\lambda}{2}\|x\|^2,$$

where $\|\cdot\|$ is the Euclidean norm and $\lambda > 0$ is a regularization parameter. The analysis for general $\lambda$-strongly convex regularizers is left for future work. For the squared Euclidean norm we have

$$g^*(\alpha) = \frac{1}{2\lambda}\|\alpha\|^2 \qquad \text{and} \qquad \nabla g^*(\alpha) = \frac{1}{\lambda}\alpha \ .$$

We further assume that each $\phi_i$ is $1/\gamma$-smooth with respect to $\|\cdot\|$, namely,

$$\forall x, z, \quad \phi_i(x) \le \phi_i(z) + \nabla\phi_i(z)^\top (x - z) + \frac{1}{2\gamma}\|x - z\|^2.$$

For example, if $\phi_i(x) = (x^\top v_i - y_i)^2$, then it is $\|v_i\|^2$-smooth.

The smoothness of $\phi_i$ also implies that $\phi_i^*(\alpha)$ is $\gamma$-strongly convex:

$$\forall \theta \in [0,1], \ \ \phi_i^*((1-\theta)\alpha + \theta\beta) \le (1-\theta)\phi_i^*(\alpha) + \theta\phi_i^*(\beta) - \frac{\theta(1-\theta)\gamma}{2}\|\alpha - \beta\|^2.$$

We have the following result for our method.

**Theorem 1.** *Assume that $g(x) = \frac{1}{2\lambda}\|x\|_2^2$ and for each $i$, $\phi_i$ is $(1/\gamma)$-smooth w.r.t. the Euclidean norm. Suppose that the ASDCA algorithm is run with parameters $\lambda, \gamma, m, \theta$, where*

$$\theta \le \frac{1}{4}\min\left\{1 \ , \ \sqrt{\frac{\gamma\lambda n}{m}} \ , \ \gamma\lambda n \ , \ \frac{(\gamma\lambda n)^{2/3}}{m^{1/3}}\right\} \ . \tag{3}$$

*Define the dual sub-optimality by $\Delta D(\alpha) = D(\alpha^*) - D(\alpha)$, where $\alpha^*$ is the optimal dual solution, and the primal sub-optimality by $\Delta P(x) = P(x) - D(\alpha^*)$. Then,*

$$m\,\mathbb{E}\,\Delta P(x^{(t)}) + n\,\mathbb{E}\,\Delta D(\alpha^{(t)}) \le (1 - \theta m/n)^t [m\Delta P(x^{(0)}) + n\Delta D(\alpha^{(0)})].$$

*It follows that after performing*

$$t \ge \frac{n/m}{\theta}\,\log\left(\frac{m\Delta P(x^{(0)}) + n\Delta D(\alpha^{(0)})}{m\epsilon}\right)$$

*iterations, we have that $\mathbb{E}[P(x^{(t)}) - D(\alpha^{(t)})] \le \epsilon$.*

Let us now discuss the bound, assuming $\theta$ is taken to be the right-hand side of (3). The dominating factor of the bound on $t$ becomes

$$\frac{n}{m\theta} = \frac{n}{m}\cdot\max\left\{1 \ , \ \sqrt{\frac{m}{\gamma\lambda n}} \ , \ \frac{1}{\gamma\lambda n} \ , \ \frac{m^{1/3}}{(\gamma\lambda n)^{2/3}}\right\} \tag{4}$$

$$= \max\left\{\frac{n}{m} \ , \ \sqrt{\frac{n/m}{\gamma\lambda}} \ , \ \frac{1/m}{\gamma\lambda} \ , \ \frac{n^{1/3}}{(\gamma\lambda m)^{2/3}}\right\} \ . \tag{5}$$

Table 1 summarizes several interesting cases, and compares the iteration bound of ASDCA to the iteration bound of the vanilla SDCA algorithm (as analyzed in Shalev-Shwartz and Zhang [2013a])

| Algorithm | $\gamma\lambda n = \Theta(1)$ | $\gamma\lambda n = \Theta(1/m)$ | $\gamma\lambda n = \Theta(m)$ |
|-----------|-------------------|---------------------|-------------------|
| SDCA  | $n$           | $nm$        | $n$           |
| ASDCA | $n/\sqrt{m}$  | $n$         | $n/m$         |
| AGD   | $\sqrt{n}$    | $\sqrt{nm}$ | $\sqrt{n/m}$  |

Table 1: Comparison of Iteration Complexity

| Algorithm | $\gamma\lambda n = \Theta(1)$ | $\gamma\lambda n = \Theta(1/m)$ | $\gamma\lambda n = \Theta(m)$ |
|-----------|-------------------|---------------------|-------------------|
| SDCA  | $n$           | $nm$          | $n$             |
| ASDCA | $n\sqrt{m}$   | $nm$          | $n$             |
| AGD   | $n\sqrt{n}$   | $n\sqrt{nm}$  | $n\sqrt{n/m}$   |

Table 2: Comparison of Number of Examples Processed

and the Accelerated Gradient Descent (AGD) algorithm of Nesterov [2007]. In the table, we ignore constants and logarithmic factors.

As can be seen in the table, the ASDCA algorithm interpolates between SDCA and AGD. In particular, ASDCA has the same bound as SDCA when $m = 1$ and the same bound as AGD when $m = n$. Recall that the cost of each iteration of AGD scales with $n$ while the cost of each iteration of SDCA does not scale with $n$. The cost of each iteration of ASDCA scales with $m$. To compensate for the difference cost per iteration for different algorithms, we may also compare the complexity in terms of the number of examples processed (see Table 2). This is also what we will study in our empirical experiments. It should be mentioned that this comparison is meaningful in a single processor environment, but not in a parallel computing environment when multiple examples can be processed simultaneously in a minibatch. In the next section we discuss under what conditions the overall runtime of ASDCA is better than both AGD and SDCA.

## 3 Parallel Implementation

In recent years, there has been a lot of interest in implementing optimization algorithms using a parallel computing architecture (see Section 5). We now discuss how to implement AGD, SDCA, and ASDCA when having a computing machine with $s$ parallel computing nodes.

In the calculations below, we use the following facts:

- If each node holds a $d$-dimensional vector, we can compute the sum of these vectors in time $O(d\log(s))$ by applying a "tree-structure" summation (see for example the All-Reduce architecture in Agarwal et al. [2011]).

- A node can broadcast a message with $c$ bits to all other nodes in time $O(c\log^2(s))$. To see this, order nodes on the corners of the $\log_2(s)$-dimensional hypercube. Then, at each iteration, each node sends the message to its $\log(s)$ neighbors (namely, the nodes whose code word is at a hamming distance of $1$ from the node). The message between the furthest away nodes will pass after $\log(s)$ iterations. Overall, we perform $\log(s)$ iterations and each iteration requires transmitting $c\log(s)$ bits.

- All nodes can broadcast a message with $c$ bits to all other nodes in time $O(cs\log^2(s))$. To see this, simply apply the broadcasting of the different nodes mentioned above in parallel. The number of iterations will still be the same, but now, at each iteration, each node should transmit $cs$ bits to its $\log(s)$ neighbors. Therefore, it takes $O(cs\log^2(s))$ time.

For concreteness of the discussion, we consider problems in which $\phi_i(x)$ takes the form of $\ell(x^\top v_i, y_i)$, where $y_i$ is a scalar and $v_i \in \mathbb{R}^d$. This is the case in supervised learning of linear predictors (e.g. logistic regression or ridge regression). We further assume that the average number of non-zero elements of $v_i$ is $\bar{d}$. In very large-scale problems, a single machine cannot hold all of the data in its memory. However, we assume that a single node can hold a fraction of $1/s$ of the data in its memory.

Let us now discuss parallel implementations of the different algorithms starting with deterministic gradient algorithms (such as AGD). The bottleneck operation of deterministic gradient algorithms is the calculation of the gradient. In the notation mentioned above, this amounts to performing order of $n\bar{d}$ operations. If the data is distributed over $s$ computing nodes, where each node holds $n/s$ examples, we can calculate the gradient in time $O(n\bar{d}/s + d\log(s))$ as follows. First, each node calculates the gradient over its own $n/s$ examples (which takes time $O(n\bar{d}/s)$). Then, the $s$ resulting vectors in $\mathbb{R}^d$ are summed up in time $O(d\log(s))$.

Next, let us consider the SDCA algorithm. On a single computing node, it was observed that SDCA is much more efficient than deterministic gradient descent methods, since each iteration of SDCA costs only $\Theta(\bar{d})$ while each iteration of AGD costs $\Theta(n\bar{d})$. When we have $s$ nodes, for the SDCA algorithm, dividing the examples into $s$ computing nodes does not yield any speed-up. However, we can divide the *features* into the $s$ nodes (that is, each node will hold $d/s$ of the features for all of the examples). This enables the computation of $x^\top v_i$ in (expected) time of $O(\bar{d}/s + s\log^2(s))$. Indeed, node $t$ will calculate $\sum_{j \in J_t} x_j v_{i,j}$, where $J_t$ is the set of features stored in node $t$ (namely, $|J_t| = d/s$). Then, each node broadcasts the resulting scalar to all the other nodes. Note that we will obtain a speed-up over the naive implementation only if $s\log^2(s) \ll \bar{d}$.

For the ASDCA algorithm, each iteration involves the computation of the gradient over $m$ examples. We can choose to implement it by dividing the examples to the $s$ nodes (as we did for AGD) or by dividing the features into the $s$ nodes (as we did for SDCA). In the first case, the cost of each iteration is $O(m\bar{d}/s + d\log(s))$ while in the latter case, the cost of each iteration is $O(m\bar{d}/s + ms\log^2(s))$. We will choose between these two implementations based on the relation between $d, m$, and $s$.

The runtime and communication time of each iteration is summarized in the table below.

| Algorithm | partition type | runtime | communication time |
|-----------|---------------|---------|--------------------|
| SDCA | features | $\bar{d}/s$ | $s\log^2(s)$ |
| **ASDCA** | features | $\bar{d}m/s$ | $ms\log^2(s)$ |
| **ASDCA** | examples | $\bar{d}m/s$ | $d\log(s)$ |
| AGD | examples | $\bar{d}n/s$ | $d\log(s)$ |

We again see that ASDCA nicely interpolates between SDCA and AGD. In practice, it is usually the case that there is a non-negligible cost of opening communication channels between nodes. In that case, it will be better to apply the ASDCA with a value of $m$ that reflects an adequate tradeoff between the runtime of each node and the communication time. With the appropriate value of $m$ (which depends on constants like the cost of opening communication channels and sending packets of bits between nodes), ASDCA may outperform both SDCA and AGD.

## 4  Experimental Results

In this section we demonstrate how ASDCA interpolates between SDCA and AGD. All of our experiments are performed for the task of binary classification with a smooth variant of the hinge-loss (see Shalev-Shwartz and Zhang [2013a]). Specifically, let $(v_1, y_1), \ldots, (v_m, y_m)$ be a set of labeled examples, where for every $i$, $v_i \in \mathbb{R}^d$ and $y_i \in \{\pm 1\}$. Define $\phi_i(x)$ to be

$$\phi_i(x) = \begin{cases} 0 & y_i x^\top v_i > 1 \\ 1/2 - y_i x^\top v_i & y_i x^\top v_i < 0 \\ \frac{1}{2}(1 - y_i x^\top v_i)^2 & \text{o.w.} \end{cases}$$

We also set the regularization function to be $g(x) = \frac{\lambda}{2}\|x\|_2^2$ where $\lambda = 1/n$. This is the default value for the regularization parameter taken in several optimization packages.

Following Shalev-Shwartz and Zhang [2013a], the experiments were performed on three large datasets with very different feature counts and sparsity. The astro-ph dataset classifies abstracts of papers from the physics ArXiv according to whether they belong in the astro-physics section;

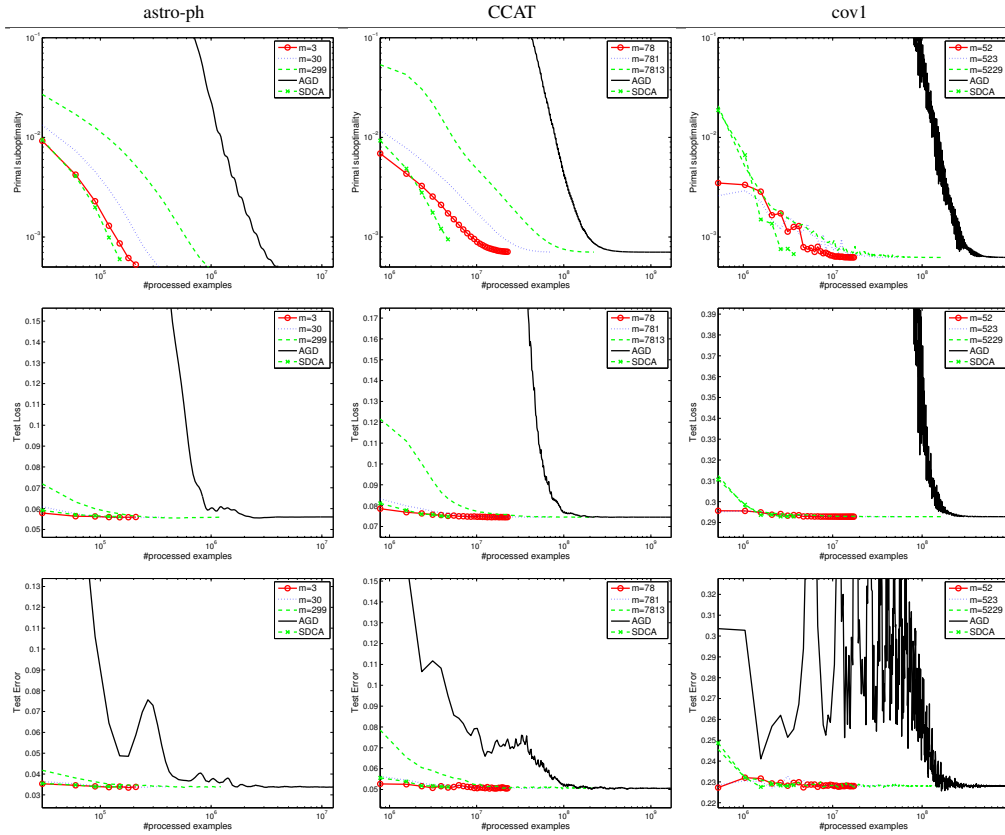

Figure 1: The figures presents the performance of AGD, SDCA, and ASDCA with different values of mini-batch size, $m$. In all figures, the x axis is the number of processed examples. The three columns are for the different datasets. Top: primal sub-optimality. Middle: average value of the smoothed hinge loss function over a test set. Bottom: average value of the 0-1 loss over a test set.

CCAT is a classification task taken from the Reuters RCV1 collection; and cov1 is class 1 of the covertype dataset of Blackard, Jock & Dean. The following table provides details of the dataset characteristics.

| Dataset | Training Size | Testing Size | Features | Sparsity |
|---------|---------------|--------------|----------|----------|
| astro-ph | 29882 | 32487 | 99757 | 0.08% |
| CCAT | 781265 | 23149 | 47236 | 0.16% |
| cov1 | 522911 | 58101 | 54 | 22.22% |

We ran ASDCA with values of $m$ from the set $\{10^{-4}n, 10^{-3}n, 10^{-2}n\}$. We also ran the SDCA algorithm and the AGD algorithm. In Figure 1 we depict the primal sub-optimality of the different algorithms as a function of the number of examples processed. Note that each iteration of SDCA processes a single example, each iteration of ASDCA processes $m$ examples, and each iteration of AGD processes $n$ examples. As can be seen from the graphs, ASDCA indeed interpolates between SDCA and AGD. It is clear from the graphs that SDCA is much better than AGD when we have a single computing node. ASDCA performance is quite similar to SDCA when $m$ is not very large. As discussed in Section 3, when we have parallel computing nodes and there is a non-negligible cost of opening communication channels between nodes, running ASDCA with an appropriate value of $m$ (which depends on constants like the cost of opening communication channels) may yield the best performance.

# 5 Discussion and Related Work

We have introduced an accelerated version of stochastic dual coordinate ascent with mini-batches. We have shown, both theoretically and empirically, that the resulting algorithm interpolates between the vanilla stochastic coordinate descent algorithm and the accelerated gradient descent algorithm.

Using mini-batches in stochastic learning has received a lot of attention in recent years. E.g. Shalev-Shwartz et al. [2007] reported experiments showing that applying small mini-batches in Stochastic Gradient Descent (SGD) decreases the required number of iterations. Dekel et al. [2012] and Agarwal and Duchi [2012] gave an analysis of SGD with mini-batches for smooth loss functions. Cotter et al. [2011] studied SGD and accelerated versions of SGD with mini-batches and Takác et al. [2013] studied SDCA with mini-batches for SVMs. Duchi et al. [2010] studied dual averaging in distributed networks as a function of spectral properties of the underlying graph. However, all of these methods have a polynomial dependence on $1/\epsilon$, while we consider the strongly convex and smooth case in which a $\log(1/\epsilon)$ rate is achievable.[2] Parallel coordinate descent has also been recently studied in Fercoq and Richtárik [2013], Richtárik and Takáč [2013].

It is interesting to note that most[3] of these papers focus on mini-batches as the method of choice for distributing SGD or SDCA, while ignoring the option to divide the data by features instead of by examples. A possible reason is the cost of opening communication sockets as discussed in Section 3.

There are various practical considerations that one should take into account when designing a practical system for distributed optimization. We refer the reader, for example, to Dekel [2010], Low et al. [2010, 2012], Agarwal et al. [2011], Niu et al. [2011].

The more general problem of distributed PAC learning has been studied recently in Daume III et al. [2012], Balcan et al. [2012]. See also Long and Servedio [2011]. In particular, they obtain algorithms with $O(\log(1/\epsilon))$ communication complexity. However, these works consider efficient algorithms only in the realizable case.

**Acknowledgements:** Shai Shalev-Shwartz is supported by the Intel Collaborative Research Institute for Computational Intelligence (ICRI-CI). Tong Zhang is supported by the following grants: NSF IIS-1016061, NSF DMS-1007527, and NSF IIS-1250985.

## Footnotes

[1]An exception is the recent analysis given in Le Roux et al. [2012] for a variant of SGD.

[2]It should be noted that one can use our results for Lipschitz functions as well by smoothing the loss function (see Nesterov [2005]). By doing so, we can interpolate between the $1/\epsilon^2$ rate of non-accelerated method and the $1/\epsilon$ rate of accelerated gradient.

[3]There are few exceptions in the context of stochastic coordinate descent *in the primal*. See for example Bradley et al. [2011], Richtárik and Takáč [2012b]

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
