[Reviews · NeurIPS 2013]

Submitted by Assigned_Reviewer_2

This paper studies a mini-batch gradient method for dual coordinate ascent. The idea is simple: at each iteration randomly pick m samples and update the gradient. The authors prove that the convergence rate of the mini-batch method interpolates between SDCA and AGD -- in certain circumstances it could be faster than both.

I am a little surprised that in case of gamma*lambda*n = O(1), the number of examples processed by ASDCA is n*\sqrt{m}, which means that in full parallelization m machines give an acceleration rate of \sqrt{m}. Since the mini-batch SGD gives the optimal m acceleration rate, I am not sure if the bound in this paper is tight. Is it possible for ASDCA to achieve the optimal rate? The experiment seems suggest ASDCA can do better than \sqrt{m}.

It will be better to compare the actual runtime of three algorithms in the empirical study. The author discusses the cost of communication and opening channels to annotate the advantage of ASDCA. However, these are not mentioned in the experiment.

In general, this is an interesting paper with nice theoretical result. A more careful discussion on the tightness of bound and a more detailed empirical study will make it better.
Summary: In general, this is an interesting paper with nice theoretical result. A more careful discussion on the tightness of bound and a more detailed empirical study will make it better.

Submitted by Assigned_Reviewer_3

The authors present an Accelerated mini-batch version of SDCA (ASDCA) and analyze its convergence rate. The theoretical results show that the ASDCA algorithm interpolates between SDCA and AGD. The authors also discuss the issue of parallel Implementation. And we observe that with an appropriate value of the batch size, ASDCA may outperform both SDCA and AGD. Some experiments are provided to support the theoretical analysis.

The paper is well written and easy to follow. The proposed algorithm is a nontrivial extension of SDCA. The algorithm is important in practice, since it can utilize parallel computing to reduce the running time. The paper is potentially impactful for the optimization and learning communities.

Some suggestions:
1. Besides the idea of mini-batches, the authors also employ Nesterov’s acceleration method in the proposed algorithm. It is not clear to me why this is necessary. What will happen if we apply the naïve mini-batching SDCA to the optimization problem considered in this study?
2. Similar to Shalev Shwartz and Zhang [2013], the linear convergence rate only holds in expectation. Is it possible to derive a high probability bound?

Typos:
1. Line 52, \gamma --> \frac{1}{\gamma}
2. Line 314, 10-2 --> 10^{-2}
Summary: The proposed ASDCA is a nontrivial extension of the existing work, and leads to a speedup in parallel computing.

Submitted by Assigned_Reviewer_5

This paper extends stochastic dual coordinate ascent (SDCA) to the mini-batch setting, with a proposed algorithm accelerated mini-batch SDCA (ASDCA). Its rate of convergence interpolates between SDCA and accelerated gradient descent (AGD). In addition, a parallel implementation is also proposed where SDCA again interpolates between SDCA and AGD. This interpolation is also demonstrated in experiment.

Although the rates of convergence for ASDCA are interesting, a careful look at Table 2 and Figure 1 reveals that ASDCA is nowhere superior to SDCA. I am wondering why it is useful to have such an algorithm. In addition, Section 3 showed much promise of ASDCA on parallel architecture. However, no experiment was done in this setting. Therefore more experimental studies (especially on parallel architecture) are necessary before we can confirm the empirical usefulness/superiority of ASDCA (as indicated by the word ‘Accelerated’ in the title).

In response to author's rebuttal: I stick to my position that the paper should show empirical evidence of the advantage of the new algorithm. I think this is an important issue, because in practice the acceleration proffered by parallelization can be highly tricky. The theory part of the paper is surely good. So I won't mind if the paper is accepted.
Summary: This paper proves the rate of convergence for a stochastic dual coordinate ascent (SDCA) algorithm in the mini-batch setting. However, no theoretical or empirical advantage is shown for this algorithm compared with SDCA. The discussion on parallel implementation also lacks experimental study.
Author Feedback

Author rebuttal: To reviewers 2 and 3:
===============

Thanks for the positive feedback.
Regarding optimality: This is a good point, which we’re still trying to figure out.
Regarding acceleration: While we don’t have a rigorous explanation for this, it seems that some sort of acceleration is necessary for improving the rate using mini-batches. This is also the approach taken in previous algorithms for sgd (such as [1,2]).
Regarding expectation vs. high probability: It is possible to obtain high probability bounds by a rather simple amplification argument.

To reviewer 5:
===========

Our theoretical and experimental results show that in many cases, by taking a mini-batch of size larger than 1 (but not too large), the total number of processed examples remain roughly the same. As discussed in Section 3, this immediately implies a potential significant improvement when using a parallel machine (such as a cluster or a GPU). This is the type of theory exist in the literature for mini-batching --- for sgd, minibatch also showed no improvement compared to sgd in the serial computing setting (such as [1,2]). However these paper are still important in the parallel setting. In this sense, your statement that “no theoretical advantage is shown” is not quite accurate.

Regarding the experiments:
Performing an experiment with a parallel machine involves many more implementation details, which are beyond the scope of a theoretically oriented NIPS submission. Our contribution is mainly theoretical, and the experiments come to validate our theoretical findings.
As a side note, most previous papers (many of them appeared at NIPS), follow the exact experimental evaluation method that we used (e.g. [1,3,4]).

[1] Andrew Cotter, Ohad Shamir, Nathan Srebro, and Karthik Sridharan. Better mini-batch algorithms via accelerated gradient methods. NIPS, 2011.

[2] Ofer Dekel, Ran Gilad-Bachrach, Ohad Shamir, and Lin Xiao. Optimal distributed online prediction using mini-batches. JMLR, 2012.

[3] Takac et al. Mini-batch primal and dual methods for SVM. JMLR 2013.

[4] Alekh Agarwal and John Duchi. Distributed delayed stochastic optimization. CDC 2012